# Managing Compound Hazards: Impact of COVID-19 and Cases of Adaptive Governance during the 2020 Kumamoto Flood in Japan

**DOI:** 10.3390/ijerph19031188

**Published:** 2022-01-21

**Authors:** Takako Izumi, Sangita Das, Miwa Abe, Rajib Shaw

**Affiliations:** 1International Research Institute of Disaster Science (IRIDeS), Tohoku University, Sendai 9808572, Japan; 2Independent Consultant, Sagamihara City 2520303, Japan; sangita.r.das@gmail.com; 3Kumamoto Innovative Development Organization (KIDO), Kumamoto University, Kumamoto 8608555, Japan; m-abe@kumamoto-u.ac.jp; 4Graduate School of Media and Governance, Keio University, Tokyo 1088345, Japan; shaw@sfc.keio.ac.jp

**Keywords:** COVID-19, flood, compound hazards, evacuation, volunteerism, adaptive governance

## Abstract

Japan experienced natural hazards during the COVID-19 pandemic as some other countries did. Kumamoto and Kagoshima prefectures, including many other parts of southern Japan, experienced record-breaking heavy rain on 4th July 2020. While many countries were affected by compound hazards, some cases such as the Kumamoto flood did not cause a spike of the COVID-19 cases even after going through massive evacuation actions. This study aims to understand how COVID-19 made an impact on people’s response actions, learn the challenges and problems during the response and recovery phases, and identify any innovative actions and efforts to overcome various restrictions and challenges through a questionnaire survey and interviews with the affected people. With an increase in the risk of compound hazards, it has become important to take a new, innovative, and non-traditional approach. Proper understanding and application of adaptive governance can make it possible to come up with a solution that can work directly on the complex challenges during disasters. This study identified that a spike of COVID-19 cases after the disaster could be avoided due to various preventive measures taken at the evacuation centers. It shows that it is possible to manage compound hazard risks with effective preparedness. Furthermore, during emergencies, public-private-partnership as well as collaboration among private organizations and local business networks are extremely important. These collaborations generate a new approach, mechanism and platform to tackle unprecedented challenges.

## 1. Introduction

Compound hazards that combined natural hazards and the COVID-19 pandemic have had major impacts on the community and the environment, and consequently increased the virus spread [1,2]. In such multiple-hazard crises, governments and other responding agents are required to make complex, highly compromised, hierarchical decisions aimed to balance COVID-19 risks and protocols with disaster response and recovery operations. For example, the aggregation of evacuees into communal environments and increased demand on medical, economic, and infrastructural capacity associated with natural hazard impacts also increases COVID-19 exposure risks and vulnerabilities [3]. In India and Bangladesh, the number of COVID-19 cases drastically increased after the evacuation of more than 6.5 million, while the timely evacuation limited mortality as Cyclone Amphan affected both countries in May 2020 [4].

As in the case of Cyclone Amphan during COVID-19, two extreme events that are not related in origin but occur simultaneously or in succession are considered compound, multiple, or concurrent hazards [5]. They also have a characteristic of amplifying the impacts with the combined events and causing an extreme event when combined [6]. These compound hazards make the preparedness and response efforts more complex and challenging for both hazards. Protocols to combat the COVID-19 pandemic include the practice of social distancing and self-isolation; however, the emergency response such as evacuation actions requires expanded planning efforts for evacuation and sheltering options to limit possible virus exposure to evacuees and essential personnel [7].

Japan also experienced natural hazards during the COVID-19 pandemic. Kumamoto and Kagoshima prefectures of southern Japan experienced record-breaking heavy rain on July 4th, 2020. The rain caused devastating floods and landslides in many areas, which killed 83 people, and destroyed 15,335 buildings according to the Fire and Disaster Management Agency (FDMA; data available online at https://www.fdma.go.jp/disaster/info/items/200709_ooame26.pdf; accessed on 16 October 2020). Sixty-five people died in Kumamoto prefecture alone. Kumamoto was the prefecture that was hit by a massive earthquake of 7.0 magnitude in 2016, which, according to an experienced volunteer who answered during an interview, prepared the prefecture in many aspects of disaster management compared to some other parts of Japan. However, the flood of July 2020 presented a number of new challenges because of the ongoing fight against the pandemic. The first severe COVID-19 positive patient was confirmed in Kumamoto Prefecture on 21 February 2020 (source: Kumamoto prefecture’s website: https://www.city.kumamoto.jp/hpkiji/pub/Detail.aspx?c_id=5&id=36067; accessed on 16 October 2020). Prior to the torrential rain disaster on 4 July 2020, no positive patients had been identified in the municipalities in the southern region of Kumamoto Prefecture, such as Hitoyoshi and Yatsushiro, which were affected by the torrential rain. Surveys through interviews of residents’ actions immediately after the disaster revealed a low level of alertness to infectious diseases in the early stages of public shelters when people were evacuated to temples, public halls, and elementary school gymnasiums.

Although at the peak 2512 people were evacuated on July 12, it did not cause a drastic increase in the COVID-19 cases resulting from close contacts at evacuation centers [8]. As of July 14, only 49 infected persons had been reported in Kumamoto Prefecture, and no infections were confirmed in the southern region of the Prefecture, including the areas affected by the torrential rain (according to the data published on the website of Kumamoto prefecture; available at https://www.pref.kumamoto.jp/uploaded/attachment/112125.pdf; accessed on 16 October 2020) Tashiro and Shaw [9] argued that this initial success could be related to the area’s culture (such as non-touch greeting and sanitation practice since childhood), food habits, and advance healthcare system. However, Japan has been managing the response to COVID-19 pandemic under a “Special Measures Act to Counter New Types of Influenza” and not the “Disaster Countermeasure Basic Act” that covers natural disasters [10]. Although efforts were made to tackle this unique situation through emergency revision of the management guidelines, both the preparation for and the response to the July 2020 flood were eventually affected by the pandemic in many ways [11]. The evacuation centers in Kumamoto took various infection preventive measures against COVID-19 such as hand disinfection, body temperature checks, zoning of people with fever, etc. [12]. Social distancing was strictly maintained among the evacuees, which reduced the usual capacity of the designated evacuation centers to a great extent [11].

While many countries were affected by compound hazards and were forced to take extra measures and efforts to tackle different types of hazards at the same time, some cases such as the Kumamoto flood did not cause a spike of the COVID-19 cases, even after going through massive evacuation actions. The Cabinet Office and Fire and Disaster Management Agency (FDMA) in Japan jointly announced the general principle in evacuation that “people in dangerous places must be evacuated in the event of a disaster, and that this principle applies during the ongoing COVID-19 pandemic”. Such an announcement and recommendation were made explicitly to prevent people from avoiding evacuation out of pandemic-related fears. At the same time, people were recommended and encouraged to evacuate to relatives’ and acquaintances’ places in addition to public shelters, or choose places that were not likely to be flooded, sleep in a car that was parked on a safe ground, and even move to the upper floor at home. This method is referred to as “dispersed evacuation” [13].

This study aims to understand how COVID-19 made an impact on people’s response actions, learn the challenges and problems during the response and recovery phases, and identify any innovative actions and efforts to overcome various restrictions and challenges through a questionnaire survey and interviews with the affected people.

## 2. COVID-19 and Flood Response: The Need for Policy Integration of Compound Risks

Based on the experience of compound hazard management under the COVID-19 situation, some research has emphasized the need for new policies and approaches to compound hazard management. Kruczkiewicz et al. [14] stressed that the existing frameworks and guidelines do not apply to compound hazards, therefore, it is crucial to redesign the institutional regulations and structures including the funding mechanism to address compound risks. Ishiwatari et al. [10] argued that new approaches are needed to respond to floods more effectively under the pandemic and future compound hazards. The new approaches should include engaging local organizations and communities, strengthening risk communication with scientific knowledge, and coordinating multiple sectors. Yusuf et al. [7] also highlighted that the pandemic-related factors should be incorporated into emergency management policies and practices. Given the compound risks include not only flood but also wild-fires, earthquakes, drought, food security, and rising temperature, various stakeholders need to cooperate and address these multiple-risks, and prepare for the increase in compound pandemic–hazard threats. Simonovic et al. [15] discussed that the new approach has to focus more on disaster resilience, which can be a rather proactive and positive approach, as well as action-based resilience planning, rather than focusing on one hazard at a time. It is also vital to understand people’s behavior to communicate what is resilience and how to prepare for and respond to these complicated events.

A literature search on the July 2020 Kumamoto flood was performed using CiNii (Citation Information by National Institute of Information) on September 15th with keywords “Disasters” and “COVID-19”, and it generated 142 results. However, few results were yielded when the search was restricted to “the 2020 Kumamoto flood”, which was the first natural hazard that Japan faced since the beginning of the COVID-19 pandemic. Only eight articles were generated by searching “the 2020 Kumamoto flood” and “COVID-19”. Very few researches on the impact, linkage, and relation between the flood and COVID-19 in Japan were made.

Uchiyama and Danjo [16] addressed the effectiveness of a hazard map that showed the areas expected to be inundated by the Kumamoto flood. It proved that a hazard map could contribute specifically to developing an evacuation plan and drill. It also pointed out the difficulty of gaining volunteers under COVID-19 and that this delayed the recovery efforts in Kumamoto. Kawata [17] emphasized the need for transforming the current focus on disaster risk management and including the risk management from the cultural perspective, which considers human lives, culture, customs, and behaviors, not only the focus on the infrastructure and urban/city planning measures to prepare for a compound hazard risk. This is called “cultural disaster risk reduction”. As such, a number of studies highlighted the need for transforming the current risk management approaches to new ones; however, the challenge is how the new approach should look and be developed. For that purpose, it is crucial to collect the case studies of responses to compound hazards and analyze them—what is missing and what went well, as well as how they could be widely applied. This study aims to showcase the local initiatives taken to overcome the challenges and continue the response and recovery efforts by local people without volunteers from outside of the affected prefecture, as well as the evacuation center management that avoided spreading the virus and keeping a safe environment for evacuees. These initiatives and approaches are considered “adaptive governance (AG)”. At the end, this study highlights the importance of its concept as one of the potential approaches in a new strategy towards compound hazard response and recovery.

## 3. The Context of Frequent Flooding in Kumamoto and Attempts to Control Them

The Kuma River basin in Kumamoto prefecture is prone to flooding almost every time there is heavy rainfall in the region (Table 1). A class-A river (a first class river designated by the Ministry of Land, Infrastructure, Transportation and Tourism in Japan, indicating rivers that are especially important to the national economy) of 115-km length, Kuma River’s course begins in the mountain range in Kyushu. After running through Hitoyoshi city, Kuma village, and Yatsushiro city of Kumamoto prefecture with a strong current, it discharges into the Yatsushiro Sea. Due to its location in South Japan, where heavy rain is very common, severe flooding episodes along the Kuma River have happened many times in recorded history, and the worst of these were in the mid-60s. When the area was affected by severe flooding three years in a row from 1963, the Construction Ministry (a predecessor of the Ministry of Land, Infrastructure, Transport and Tourism or MLIT in short) decided to construct a dam along the largest arm of the Kuma River as a flood control measure. The proposed dam, however, could have potentially affected the water flow, which eventually could have affected the tourism industry and agriculture. The central government, therefore, canceled the project in 2009 following resistance from the prefecture.

Three months after the July 2020 flood, an estimation compiled by the MLIT (available online: http://www.qsr.mlit.go.jp/yatusiro/site_files/file/bousai/gouukensho/20201006shiryou2.pdf; accessed on 5 August 2021) was revealed, which showed that the canceled dam could have reduced the total area of inundation by 60.7%. The prefectural government has started to reconsider the proposed dam with design modifications following this revelation [18].

## 4. The Survey: Method and Key Findings

The online survey was conducted from March 2nd to March 11th in 2021, with the support of a survey company, to understand the impact of COVID-19 on the evacuation action taken by the residents and the change in volunteerism under the Kumamoto flood, and identify innovative response and recovery efforts taken by local communities to tackle the difficulties that they encountered during COVID-19 and the flood.

The questionnaire was constructed with a major focus on (1) the impacts of COVID-19 and the flood, (2) evacuation, and (3) volunteerism. The questionnaire was distributed and the respondents were identified by a survey company through their local networks. The constraint of this survey was that only people who have internet access were able to participate in this survey. A total of 276 samples were collected from the people in seven cities/villages of Kumamoto prefecture that were affected by the flood of July 2020. The data collected by the survey were compiled, digitized, analyzed using Microsoft Excel.

A breakdown of the target areas is presented in Table 2. The gender ratio of men and women was equal in this survey. The percentage of respondents in their 20s and 30s, 40s and 50s, and 60s and 70s were 53%, 36%, and 11%, respectively.

In addition to the questionnaire survey, interviews and hearings with some volunteers and local affected people were conducted from July to October in 2020 to understand the response practices made by the evacuation management committee and the volunteers. The interviews were also conducted with nonprofit organizations (NPOs) including Young Men’s Christian Association (YMCA) and Peace Boat, who managed the evacuation centers at that time. Affected business owners who established a volunteer base, and local volunteers including students and business persons such as bankers were also interviewed. Through these interviews and hearings, innovative strategies and mechanisms on not spreading the virus at the evacuation centers, and ensuring human resources to assist the response and recovery efforts, were identified and introduced in this paper.

### 4.1. Impacts of COVID-19 and the Flood

Ninety-three percent of the respondents reported that the pandemic impacted their lives. The negative impacts on mental health, income, and social ties were relatively high (Figure 1).

In contrast to COVID-19, only 55% of the respondents answered that the flood impacted their lives. The largest impact was on mental health, and there was a wide gap between this and second largest impact, which was on income (Figure 1). Regardless of disaster type (pandemic or natural disaster), both COVID-19 and the flood had a serious influence on mental health in particular.

### 4.2. Evacuation

Of the 276 respondents, 43 (16%) evacuated before the flood. Because heavy rains occurred from midnight to early morning, it may have been difficult to evacuate during that period of time. Among the respondents who evacuated, 80% answered that the risk of COVID-19 infection impacted their decision to evacuate. The greatest impact was that it took time to make the decision to evacuate.

Thirty percent of those who evacuated went to other places perceived to be safer instead of going to the designated evacuation center. Among them, 67% evacuated to a car and 25% evacuated to the home of a friend or someone they knew. These locations might have been chosen because of the evacuation situation at the time of the Kumamoto earthquake in 2016.

The respondents reported that “access to food and water” was their biggest concern when deciding whether to evacuate to an evacuation center (Figure 2). Their second most important concern was exposure to COVID-19 and “caring for children”. However, the options included three COVID-19 related concerns—“COVID-19 infection”, “cannot take enough preventive measures against COVID-19”, and “cannot enforce sufficient social distance at the center”. When these numbers are summed up, COVID-19 related concerns appear to be as important as “access to food and water”. These concerns most likely stem from prior experience at an evacuation center or their understanding of the general conditions at the centers. Many people hold negative images of evacuation centers.

The major problems encountered at the evacuation centers were “not enough food and other necessary items” (a major concern of evacuees regarding the evacuation centers before their evacuation) and “lack of privacy” (Figure 3). These problems were not related to COVID-19 but were rather common problems encountered at evacuation centers in Japan. In addition, around 20% of the respondents considered “using public toilets, washrooms, and baths” and “having to wear a mask all day” to be major problems. Only 12% considered having some social distancing at the evacuation centers to be a major problem. This finding indicates that the infection prevention measures, especially the practice of social distancing, had been taken at most of the evacuation centers; thus, people did not have to worry about it.

With regard to the reason why the respondents did not evacuate, 74% answered that they did not evacuate because they thought it was possible to secure their safety without evacuation. This indicates that 26% of those who did not evacuate considered not evacuating unsafe, but, for various reasons, still hesitated to evacuate. Improving or solving the problems at the evacuation centers may reduce this hesitation to evacuate and, thus, encourage more people to take necessary action as quickly as possible. Concerns about COVID-19 infection was the second most cited reason (9.4%) for not evacuating, which shows that the pandemic did affect people’s decision to evacuate. Only 1% thought that the evacuation itself was too much of a hassle.

### 4.3. Volunteerism

Among the 276 respondents, 11% (30) received support from volunteers. The small percentage was primarily due to the shortage of volunteers because of COVID-19-related restrictions. For instance, the volunteers were limited to Kumamoto prefecture to avoid spreading the virus (Website of Social Welfare Council https://www.saigaivc.com/202007/; accessed on 2 November 2020).

Volunteers provided different types of support during and after the disaster (Figure 4). During the first week, the need was mostly to assist in distributing food, provide emergency items, and cleaning houses. After the first week, the need shifted to cleaning houses and providing urgent mental health support. After about a month, the focus was distributed to various types of support at the same time.

Twenty-three percent of the respondents participated in volunteer activities. However, 77% (214) did not participate in any volunteer activities. The most cited reason for not participating was that they were “worried about COVID-19”, which again shows a link between the pandemic and the shortage of volunteers. Fifteen percent did not participate in any volunteer activities because they were also affected by the flood.

## 5. Discussion with Case Studies of Evacuation and Volunteerism

The survey findings showed that COVID-19 and the flood had severe negative impact on mental health, income, and social ties. In addition, the compound disaster caused by COVID-19 and the floods had a significant influence on the areas of evacuation and volunteerism. This section further focuses on these two areas, introducing case studies collected mainly through interviews and observations at the time of the flood.

### 5.1. Evacuation

The survey findings revealed that COVID-19 had a major impact on the decision to evacuate. Respondents took more time to make the decision and many evacuated to places other than a designated facility. Thirty percent of the evacuees went to other places out of fear that they or their family members could be exposed to COVID-19.

Evacuation centers also had to take various preventive measures to prevent further spread of infection [13,19], following existing guidelines and the newly issued instructions from the government. The WHO [20] has pointed out that it is crucial that all preventive measures are carried out at evacuation centers, such as hand hygiene, wearing masks, social distancing, etc. It was also strongly recommended by the Japanese government to avoid the “three C’s” that is, “closed spaces, crowded places, and close-contact settings”, in the response to disasters in the midst of the pandemic [13]. The evacuation process and its management during COVID-19 and the flood were more complicated and challenging than the management of a single crisis. Some of these special measures taken in Kumamoto are described below.

#### 5.1.1. Controlled Entrance and Setting Hand Sanitizers and Thermography

The entrance and exit at the evacuation centers were strictly controlled. The designated evacuation centers, which are usually gymnasiums of public schools, have more than one door, through which people can come and go according to their needs. However, in 2020, only one door was used for both the entrance and exit, so that only those who had permission could enter after a temperature check and hand sanitizing. A thermograph was placed in the reception area (Figure 5), as well as bottles of hand sanitizers.

While it is not uncommon to have hand sanitizers at evacuation centers, their placement at the entrance as well as at the common spaces, with monitoring by the management to ensure their proper use, was a measure taken for the first time. In most evacuation centers, hand sanitizers were placed in areas such as mobile phone charging stations (Figure 6) and library corners, apart from the reception at the entrance.

#### 5.1.2. Social Distancing within the Evacuation Centers

In addition to setting preventive items, social distancing is also an effective measure for preventing viral transmission [21]. However, an evacuation center is often overcrowded and there is a high risk of COVID-19 infection [19,22]. To ensure adequate social distancing in order to prevent the spread of the virus, the evacuees were asked to stay only within a makeshift designated area marked by light materials such as cardboard or plastic floor mats. In some evacuation centers, the evacuees were dispersed throughout the facility, including available classrooms.

#### 5.1.3. Dispersed Evacuation

The social distancing measures described above reduced the capacity of an evacuation center to nearly 30%. It was therefore crucial to designate more evacuation centers to accommodate the remaining evacuees, and, also, identify isolation facilities for infected individuals [23]. In order to increase the number of evacuation centers, non-traditional and non-designated evacuation centers such as hotels, other accommodation facilities, training facilities, and other potential places for evacuation were identified and people were encouraged to use them. This method is referred to as “dispersed evacuation” [13]. To implement “dispersed evacuation”, municipalities became responsible for securing a sufficient number of designated evacuation centers in order to prevent secondary disasters.

Furthermore, it is also crucial that the evacuation of people to neighboring towns and cities through advanced agreements among neighboring municipalities is considered in case there is a shortage of evacuation centers. As of July 9, 2020, there were 403 evacuees from Kuma village in Kumamoto Prefecture, of whom it was only possible to accommodate 131 at four designated evacuation centers in Kuma village. Because the number of designated centers was insufficient, 272 evacuees were accommodated at the centers outside the village with support from neighboring municipalities [13]. Such efforts and initiatives to carry out evacuations to wider areas, beyond a traditional evacuation pattern that includes only designated centers in the evacuee’s residential village, can expand the capacity to cope with large-scale disasters.

#### 5.1.4. Some Issues That Call for More Attention

The preventive measures described above had some negative effects too. For instance, the measures limited exchanges among volunteers and evacuees, especially older adults who were assigned in small numbers to small classrooms. There was very limited opportunity for conversations or brief exchanges between evacuees and volunteers. The primary focus of all the extra measures was to prevent the spread of COVID-19 and keep the evacuees protected till they could be moved to individual temporary houses. This, in turn, may have forced some of the evacuees to keep their problems and discomfort to themselves, blocking the way for timely solutions.

Another important issue that came to light in the recent years was the environment at the evacuation centers. In June 2013, based on the experience of the Great East Japan Earthquake and Tsunami in 2011, Japan’s Basic Act on Disaster Countermeasures was amended, and “Securing a good living environment at evacuation centers” was added. It became a responsibility of municipalities for securing a certain level of living environment at the centers, not only securing sufficient number of evacuation centers [24]. However, the issues such as “improvement of environment at evacuation centers” and “divergence of evacuation places” have been left unsolved until present [25]. Murosaki (2020) has pointed out that a general negative image about evacuation centers being “distant, unclean, and no-space” often causes the hesitation to evacuate. These issues have existed for many years, and the COVID-19 situation exposed them further. It is urgently required to review these issues and take actions to improve the situation [26].

### 5.2. Volunteerism

The survey findings indicated that a very limited number of people (11%) received assistance from volunteers. This finding can be explained by the severe shortage of volunteers due to travel restrictions during COVID-19. The first respondents during the initial weeks were individuals and organizations from within the prefecture [11]. The survey also showed that people did not participate in volunteer activities because they were “concerned about the possibility of COVID-19 infection” (57%). Only 15% answered that it was because they were also affected by floods.

Volunteer support is crucial in disaster response [25,27]. Volunteers provide a significant resource for emergency management [28]. There are several types of volunteers. Waldman et al. [29] categorized them into spontaneous volunteers and affiliated volunteers associated with emergency-related voluntary service organizations. Spontaneous volunteers participate in volunteer activities from outside the affected areas and do not have any formal training in disaster response [30]. If they do not have any formal training and are from the same community or neighboring areas, they are called “informal volunteers” who work outside of formal emergency and disaster management arrangements [30]. The communication and coordination among spontaneous or informal volunteers and affiliated volunteers are ineffective, and affiliated volunteers have concerns about the level of safety and liability associated with the work of spontaneous volunteers [28]. Whittaker et al. [28] indicated that those who work outside of systems tend to be viewed as nuisances or liabilities, and their efforts are often undervalued.

An “on-call civilian firefighter” is an example of a volunteer who has been trained. If they are from outside of the affected area, they are “expert volunteers” (e.g., health workers) or “formal or affiliated volunteers”. Because disaster risks are increasing, it is expected that the need for and expectation of spontaneous volunteers will also increase [31]. It has become necessary to estimate the minimum manpower required for the cleaning and reconstruction work at each disaster-prone area, and create a sort of roadmap towards ensuring spontaneous volunteers from within the area itself [11].

During the Hanshin-Awaji earthquake disaster in Japan in 1995, a large number of volunteers came to the affected area to provide assistance; however, it was not possible to understand the needs of the affected people and effectively coordinate responses. Based on the lessons learned from these experiences, disaster volunteer centers (VCs) that aim to understand the needs of the affected people and coordinate the activities of volunteers are being established by the Social Welfare Association in the affected areas in collaboration with nonprofit organizations (NPOs) and volunteer organizations [32,33]. In addition, an NPO support center is often established to share tasks with VCs. While an NPO center collaborates with VCs, there is no clear division of work and responsibilities between VCs and NPO support centers [34]. In the case of Japan, most of the volunteers are usually spontaneous (coming from outside the affected areas/no formal training) or informal (coming from the same communities or neighboring areas/no formal training); however, they normally work after being registered as volunteers at disaster volunteer centers.

COVID-19 prevented volunteers from participating in support activities such as helping to clean houses and removing rubble, distributing necessary items, and cooking and distributing food [35] as these activities can interrupt social distancing [11]. In the case of the flood in Kumamoto in 2020, the affected people were allowed to accept support only from local volunteers (https://www.saigaivc.com/202007/); accessed on 2 November 2020). The challenge of how to ensure the necessary support previously given by volunteers for relief and recovery assistance has emerged during the pandemic. Through interviews with affected people and volunteers, this study identified four new strategies for increasing volunteer support, in order to overcome this challenge.

#### 5.2.1. Establishment of Volunteer Base by Business Owners

The recruitment of spontaneous volunteers was limited to those within Kumamoto prefecture, which did not ensure sufficient support personnel for the restoration of even ordinary homes. Therefore, restoration activities for hotels, stores, shops, etc., in the center of the city were aided not by volunteers dispatched from volunteer centers but by relatives, acquaintances, and those who in normal times had been business partners of the shop owners. Within Hitoyoshi City, private volunteer bases were set up, and stores performing restoration work began to appear and function as new hubs of activity. One man established a private volunteer group at his store’s site. He manages a wholesale store selling a distilled drink from Kumamoto. This one-storied shop was flooded up to the ceiling. The center welcomed supporters from partner businesses, banks, etc., and placed containers at the site. The volunteers removed debris from disaster-struck stores, cleared away mud, and cleaned up.

At the volunteer base, drinks and ice were provided, a rest area was set up, and a large number of electric fans were operated to prevent heatstroke from working in the blazing sun. Due to the shortage of volunteers because of reasons described in the previous sections, there was very little expectation of volunteers being dispatched from official volunteer centers. Therefore, this private volunteer base was set up by a storeowner so that the efforts could be carried out by themselves. The volunteers were requested to wear masks, wash hand, use disinfections frequently, and measure their temperature in order to pay close attention to not spreading the virus. Moreover, the information of who worked where and with whom were saved in case someone became infected.

#### 5.2.2. Recruiting Support Staff under “The Kuma Recovery Project”

To deal with the shortage of volunteers, a group of local organizations, in cooperation with the business community and other stakeholders, started a project called “The Kuma Recovery Project” a month after the 2020 flooding. Their objective was to connect three parties: (1) areas where volunteers were needed for cleaning work, (2) potential volunteers, who were mostly people who lost their business or jobs to the disaster, (3) contributors and fund providers.

There were certain criteria for people who could apply as volunteers. The potential volunteers could apply through the website, where a list of work sites and dates were posted. Based on the date and place of work, bus rides were provided. After a day’s work, the volunteers were given a small amount of daily allowance from the fund created by the contributors. The website regularly posted updates of the recovery work performed under this project.

#### 5.2.3. Involving Local Students in Relief Activities

The experience of the flood in 2020 during COVID-19 highlighted the challenge of volunteer service operations under the pandemic and the need for unconventional approaches to gain support for disaster relief. For instance, local students in Kumamoto prefecture were recruited to replace volunteers and were paid a minimum wage to provide the services normally provided by volunteers. Recruitment was carried out by the “Kumamoto Support Team,” a private organization set up by volunteer members from Kumamoto Prefecture after the Great East Japan Earthquake and Tsunami in March 2011 to establish a support system to provide sustainable relief assistance. This organization became a general incorporated association in July 2020.

The organization used donations received from across Japan to manage volunteer activities by paying a daily allowance to university students who participated in the relief activities. Students who had lost income from part-time jobs due to COVID-19 were employed by the organization. In addition to a daily allowance, participating students were given meal coupons for use at restaurants and shops affected by the flood as part of a new initiative to help restaurants and shops affected by income loss. The initial budget was obtained through crowdfunding, with approximately JPY 20 million raised across Japan, exceeding the target amount of JPY 6 million. The total number of students who worked under this system had exceeded 1450 as of 28 September 2020 (Source: Kumamoto Support Team website: https://kumamoto-team.net/; accessed on 5 July 2021).

#### 5.2.4. Crowdfunding to Support Response and Recovery Efforts

At the time of the flood in July 2020, a private organization, “Student Disaster Volunteer Support Association” was established by university and NPO staff who had been carrying out support activities for the response and recovery effort after the Hanshin-Awaji earthquake in 1995. They started crowdfunding to gain support from student volunteers who were active within the prefectures affected by the flood because it was difficult to receive support from volunteers from outside the prefecture due to COVID-19.

These funds were mainly used to support the transportation and material costs of support activities and to cover activity expenses with the goal of promoting the proactive participation of student volunteers. However, the funds did not cover a daily allowance for the volunteers, which was different from the support given by the “Kumamoto Support Team”. Applications for participation in the volunteer activities were accepted from universities with the requirement of forming a group of three or more people, not by an individual. Fundraising through crowdfunding targeted a maximum of JPY 200,000 per activity, and a total of JPY 5,000,000 for 25 projects. Within 2 months, a total of JPY 6,869,855 was raised, and the number of supporters reached 795. As of June 2021, there have been 24 completed or ongoing projects. Through this scheme, budgets were secured through crowdfunding to contribute directly to the response and recovery activities, and to enable groups to start the activities as soon as possible, not covering a daily allowance to individuals.

#### 5.2.5. Revival of Traditional Systems

The first case of the establishment of a volunteer center by business owners succeeded in gaining volunteer support from their own business networks, without relying on the volunteers dispatched by official volunteer centers. This is considered a case of reactivation of a traditional mutual help system in communities. Other cases are new schemes to provide immediate necessary support to the affected people developed on the ground, which aided in overcoming the shortage of manpower and financial resources and fueled relief and recovery efforts.

Due to the difficulty of ensuring enough volunteers, two different types of mutual aid systems arose eventually: (1) flexible changes to disaster responses under the pandemic, not only to follow a traditional response, but rather to innovate and apply new mechanisms, exemplified by the cases implemented for operating evacuation sites and securing necessary support from the volunteers, and (2) the revival of traditional systems of mutual aid that existed for a long time in the region [35]. The experience of compound hazards in Kumamoto showcased not only new and innovative mechanisms and initiatives, but also the potential for the revival of traditional and the emergence of new mutual aid systems in the community that grew under the pressures associated with the non-arrival of volunteers on the mutual aid systems that existed before the disaster [36].

## 6. Conclusions

The findings from the interviews conducted for this study bring forth some important issues that can be crucial in the management of compound disasters of a similar nature in the future. The findings show, that while concerns for COVID-19 was the main reason behind the dilemma on whether to evacuate or not for most respondents (80% felt their decision to evacuate was impacted by COVID-19), once they were in the evacuation center, their major concerns were ensuring privacy, food, and necessary items (25.6% each). Only 14% of the respondents expressed concerns over protection against COVID-19. In other words, the findings show that COVID-19 infection was a much bigger concern before the evacuation than after, indicating some success of the measures taken at the evacuation centers. Some of these measures were instructed in the government guideline, but many were not—they came about as a response to certain issues that were created because of the complexity and uniqueness of the situation.

Similarly, concerns for COVID-19 affected the local residents’ decision to volunteer at the early recovery stage. In this study, 77% of the respondents refrained from volunteering for this reason. Spontaneous volunteers from outside the prefecture could not access the affected areas because of travel restrictions in order to control COVID-19. This unprecedented situation severely affected the recovery process in the early weeks, and the strategies described in the previous section to overcome this serious issue show the importance of a participatory multi-stakeholder platform. These new strategies, including the reactivation of a traditional system to overcome the challenge of a shortage of volunteers can be explained as “adaptive governance (AG)”. An AG approach is put forward as an alternative method of managing complex social–environmental problems including disasters [37]. AG calls for new governance systems that are “less rigid, less uniform, less prescriptive and less hierarchical, and promise a more innovative but effective way of dealing with complex environmental problems” [38]. One proposed innovation for more flexible and participatory methods of governance is through the multi-stakeholder platform, defined by Steins and Edwards [39] as: “Decision making bodies (voluntary or statutory) comprising different stakeholders who perceive the same resource management problem, realize their interdependence for solving it, and come together to agree on action strategies for solving the problem.”

With an increase in the risk of compound hazards, it has become important to take a new, innovative, and non-traditional approach. Generalized guides are often not applicable to complex situations where several issues overlap. Proper understanding and application of AG can make it possible to come up with solutions that can work directly on the complex challenges during disasters.

This study eventually leads to the following conclusions:➢The COVID-19 infection was the second common concern at evacuation centers next to gaining enough food and water before people decided to evacuate. However, the most serious problem at the evacuation centers that people actually experienced was insufficient privacy, not COVID-19-related issues. This revealed a long-term problem and concern at evacuation centers. The general conditions at the evacuation centers need to be improved urgently to motivate people to take the necessary action at an emergency. At the same time, it is important to share the information on the safety and preventive measures taken at evacuation centers to eliminate “evacuation hesitation” for concerns related to infection.➢Income loss and mental health damage were most common impacts caused by COVID-19 and the flood. Therefore, financial support and assistance to recover from mental health damage are required at the recovery phase.➢Due to a lack of volunteers, it was difficult to access the necessary support normally provided by volunteers at the response and recovery phases. However, new mechanisms and initiatives to overcome this issue have evolved. The effectiveness of such innovative approaches should be further examined, and the experience should be shared widely.➢A spike of COVID-19 cases after the disaster could be avoided due to various preventive measures taken at the evacuation centers. It shows that it is possible to manage compound hazard risks with effective preparedness including timely communication and coordination.➢The preventive measures, however, restricted the interaction among the evacuees and management personnel to a great extent. This may have forced some evacuees to keep their problems and discomforts to themselves, which could lead to various serious issues including disaster-related deaths. Provision for alternative ways of communication and interaction need to be included in the evacuation centers considering this experience.➢During emergencies, public–private partnership as well as collaboration among private organizations and local business networks are extremely important. These collaborations generate a new approach, mechanism, and platform to tackle unprecedented challenges.

In recent years, the risk landscapes of the world have been increasingly complex. As stated earlier, governments and other responding agencies are having to make compromised decisions in the midst of multiple, overlapping emergencies, due to the lack of a consolidated roadmap towards the management of compound hazards. The changing climate has been adding to the complexity of these risks, making the need for a possible roadmap increasingly urgent. AG will be a key to managing compound hazards to overcome difficulties, especially during the recovery stage, that require a longer time and effort.

This study was conducted based on the experience and case studies in Kumamoto. The response and recovery efforts towards and from compound hazards, as well as their effectiveness, differ from country to country. For instance, the mass evacuation in some countries caused a rapid increase in COVID-19 cases. There should be innovative responses and recovery actions towards COVID-19 and natural hazards specific to each country. Furthermore, different challenges and problems need to be identified under different environments. Therefore, further studies, analysis, and comparison of the effectiveness of risk management on compound hazards need to be conducted based on the practices in different countries.

## Figures and Tables

**Figure 1 ijerph-19-01188-f001:**
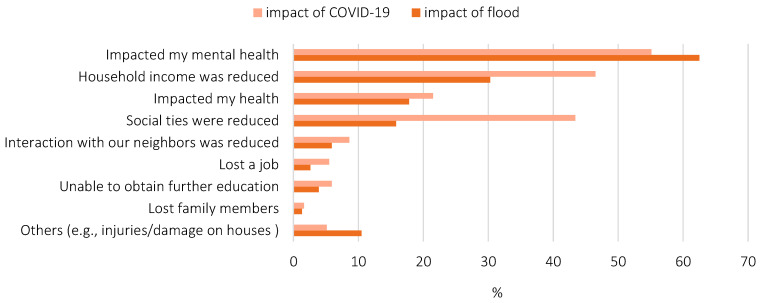
Impacts of COVID-19 and the flood.

**Figure 2 ijerph-19-01188-f002:**
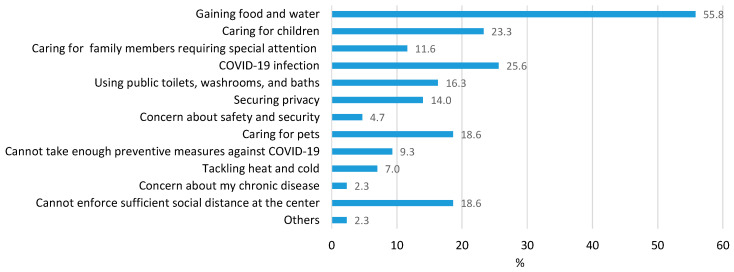
Concerns of the respondents at the time of evacuation.

**Figure 3 ijerph-19-01188-f003:**
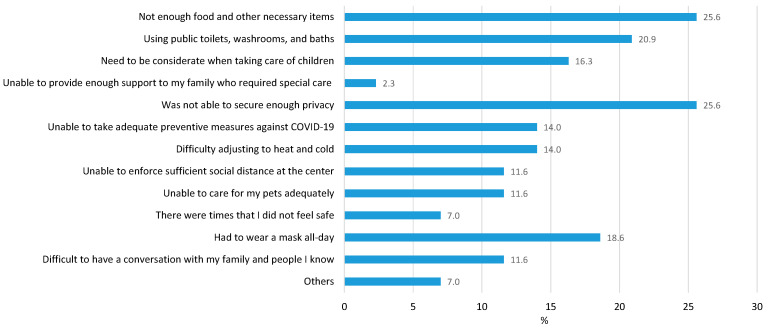
Major problems at the evacuation centers.

**Figure 4 ijerph-19-01188-f004:**
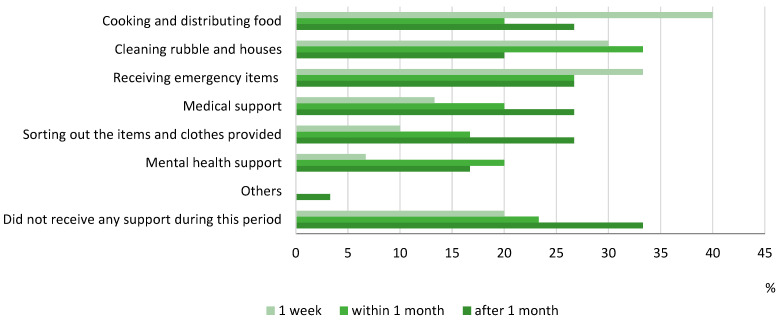
Support provided by volunteers.

**Figure 5 ijerph-19-01188-f005:**
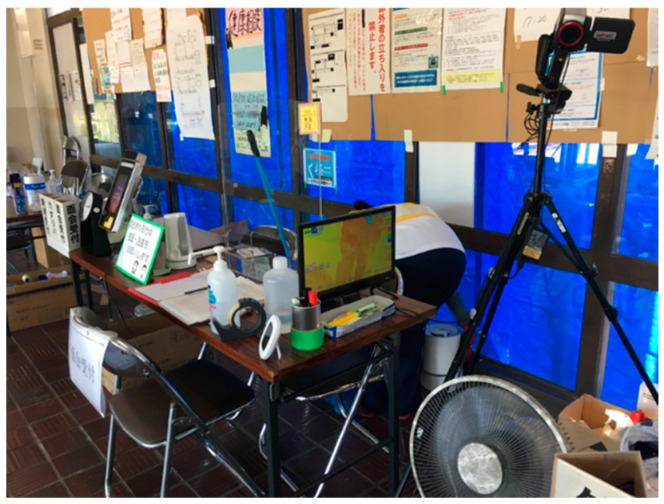
Hand sanitizer and thermography at the registration tables.

**Figure 6 ijerph-19-01188-f006:**
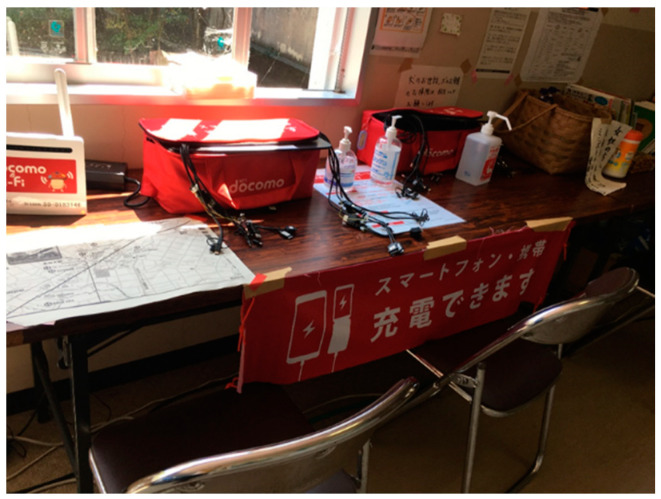
Hand sanitizers at mobile phone charging areas.

**Table 1 ijerph-19-01188-t001:** Major floods along the Kuma River since the 1960s.

July 1965	Kuma river overflowed along almost its entire length because of extremely heavy rainfall, flooding almost two-thirds of Hitoyoshi city and breaking a part of the Hagiwara levee in Yatsushiro.
July 1982	The same areas were affected along the Kuma river after a record-breaking rainfall on July 24. Over 5000 houses were inundated and 47 houses were washed away.
August 2004	Heavy rainfall (664 mm in 4 days) brought by a typhoon towards the end of August caused the Kuma river to overflow along its mid-stream, forcing people of Hitoyoshi city and surrounding areas to evacuate.
September 2005	The mid-stream of the river overflowed following heavy rainfall caused by a typhoon. In total, 119 houses were inundated and over 750 families had to evacuate.
July 2006	Continuous heavy rainfall for 5 days raised the water level all along the Kuma river, which overflowed in places inundating 80 houses. Over 900 families in Hitoyoshi city, Yatsushiro city, Kuma village, and surrounding areas had to evacuate.
June 2008	Heavy rainfall caused the Kuma river to swell and overflow inundating 33 houses. In total, 1087 families in Hitoyoshi, Yatsushiro, and Ashikita town had to evacuate.
June 2011	The water level of Kuma river crossed the danger limit after heavy rainfall (566 mm over 4 days), forcing residents of Hitoyoshi city and surrounding areas to evacuate. At least 8 houses were inundated.

(Compiled by the authors based on the data from the website of MLIT’s Yatsushiro River and National Highway Office: http://www.qsr.mlit.go.jp/yatusiro/river/kouzui/index.html; 17 September 2021).

**Table 2 ijerph-19-01188-t002:** Target areas of the survey.

Areas	No. of Answers	Percentage (%)
Kumamoto city	195	71
Yatsushiro city	43	16
Hitoyoshi city	14	5
Arao city	16	6
Tsunagi town	4	1.4
Sagara village	3	1
Kuma village	1	0.4

## Data Availability

The data presented in this study can become available on request from the corresponding author. The data are not publicly available due to legal and privacy issues.

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
