# Peer review of "Managing Compound Hazards: Impact of COVID-19 and Cases of Adaptive Governance during the 2020 Kumamoto Flood in Japan"

_ijerph, 2022, doi:10.3390/ijerph19031188_

Round 1
Reviewer 1 Report
I carefully went trough the ms by Takako Izumi et al. entitled Managing compound hazards: Impact of COVID-19 and cases of adaptive governance during the 2020 Kumamoto flood in Japan the authors submitted to International Journal of Environmental Research and Public Health. I think it is an interesting piece of work. Dealing with managment of recovery phases of a major hazard with the impending COVID-19 disease (compound hazards) is surely an interesting matter and the Kumamoto flood is a very suitable test bench. I just annotated a few concerns on the main text I hope the authors can address in the revision phase, and some typos or unclear statements that shoud be amended. Please see the attached file

Author Response
Dear Sir or Madam,
Thank you very much for the time and effort that you have dedicated to providing your valuable and insightful feedback, comments and suggestions on our manuscript. We have worked on the revision to incorporate changes to reflect the suggestions provided by you. Please find attached a point-by-point response to your comments and concerns.
We look forward to hearing from you regarding our submission and to respond to any further questions and comments you may have.
Sincerely,
Takako Izumi

Reviewer 2 Report
Thank you for the opportunity of reading and reviewing your manuscript. It addresses a topic which falls under the journal's topic. The authors present an empirical, survey-based investigation, which can bring interesting insights. However, there are several shortcomings and before publication the paper needs significant improvements.
Here are some suggestions and comments>
- the theoretical background is quite weak, there are quite limited literature references. This issue should be addressed by constructing a more solid basis for your reserach
- please clarify the research gap you identified and how exactly you can fill it through your research, which is the aim of your paper etc.
- the methodology section should clearly indicate the methods, data about the sample, how it was constructed, when the survey took place, the sample method etc.
- there is little or no indication on how your findings can be valid for larger populations and how the results can be extended for wider contexts
- in the final section you should provide not only the conclusions, but also limitations, implications, both theoretical and practical etc.
Good luck!
Author Response
Dear Sir or Madam,
Thank you very much for the time and effort that you have dedicated to providing your valuable and important feedback, comments and suggestions on our manuscript. We have tried to incorporate changes to reflect the suggestions provided by you.
Please find attached a point-by-point response to your comments and concerns.
We look forward to hearing from you in due time regarding our submission and to respond to any further questions and comments you may have.
Sincerely,
Takako Izumi

Round 2
Reviewer 2 Report
Thank you for providing a revised version of your article. It addresses most of my concerns and I consider it a better version.